# Sustainability Evaluation of Hybrid Agriculture-Tractor Powertrains

**Simone Pascuzzi** [1], **Katarzyna Łyp-Wrońska** [2], **Katarzyna Gdowska** [3,*] **and Francesco Paciolla** [4]

1   Department of Soil, Plant and Food Science, University of Bari Aldo Moro, Via Amendola 165/A, 70126 Bari, Italy; simone.pascuzzi@uniba.it
2   Department of Materials Science and Non-Ferrous Metal Engineering, Faculty of Non-Ferrous Metals, AGH University of Krakow, al. Mickiewicza 30, 30-059 Kraków, Poland; klyp@agh.edu.pl
3   Faculty of Management, AGH University of Krakow, al. Mickiewicza 30, 30-059 Krakow, Poland
4   Department of Electrical and Information Engineering (DEI), Polytechnic University of Bari, Via Edoardo Orabona 4, 70126 Bari, Italy; francesco.paciolla@poliba.it
*   Correspondence: kgdowska@agh.edu.pl; Tel.: +48-12-617-3992

**Abstract:** Agricultural tractors are highly fuel-consuming and soil/air polluting machines; thus, the introduction of new sustainable technologies, such as hybridization, can be very impactful for the development of electric hybrid agricultural tractors. These vehicles combine the classic internal combustion engine with an electric machine. This paper reports the modeling and simulation, conducted using a simulation software typically used for on-road vehicles, of a two-wheel-drive agricultural tractor in three different configurations: the conventional one, and the series and parallel electric-hybrid powertrains. The simulated task is the trailing of a "big square baler" during the process of straw wrapping and baling. The evaluation and the comparison of the fuel consumption, $CO_2$ emissions and the depth of discharge of the different configurations have been carried out to determine if it is possible to downsize the ICE while maintaining the same performance levels. This study highlights the fact that both the fuel consumption and the $CO_2$ emissions of series and parallel electric-hybrid agricultural tractors are ten times lower and five times lower than those of a traditional tractor, respectively. Furthermore, the presence of an electric machine allows a more precise speed profile tracking. This study points out that the hybridization of agricultural tractor powertrains is one of the most promising approaches for reducing pollutant emissions and fuel consumption.

**Keywords:** hybrid agricultural tractors; hybrid powertrains; fuel consumption; pollutant emissions; sustainability

## 1. Introduction

The extensive use of hydrocarbon fuels, which are employed to power a variety of agricultural machinery, including tractors and harvesters, adversely effects the environment and significantly reduces air quality [1]. Agricultural tractors are the most fuel-consuming and soil/air polluting farming machines. Furthermore, all the moving parts of the engine and of the drive train, such as piston rings, valve guides, camshafts, and transmission gearing, are under the influence of friction, which significantly contributes to mechanical losses. These losses account for 10–15% of the energy generated by the internal combustion engine (ICE) [2]. The reductions of these losses contribute to the improvement of environmental sustainability and are achieved, in the main, thanks to the appropriate choice of motor oil, which is also fundamental for ensuring the best reliability of the agricultural tractor engine [3]. The agricultural sector accounts for almost 10% of the annual EU27 production of greenhouse gasses (GHGs) [4,5]. Elevated levels of carbon dioxide ($CO_2$) emissions are considered to be one of the causes of global warming [6]; each liter of burned diesel fuel produces as much as 2.7 kg of $CO_2$ [7]. Vehicles with ICEs are the major contributor to pollutant emissions. In the agricultural sector, diesel engines represent the most widely

adopted powertrain [8]. In the literature, most of the studies present results relevant to fuel consumption when some bio-based products have been added [9–11], or evaluate the emissions of agricultural machines while fixing the ICE parameters [12–14]. In real-world field operations, these parameters usually change their values; a few studies analyze the performance levels of tractors during the execution of some real agricultural tasks [15–17], but there are no reference values relevant to the pollutant emissions generated.

The monitoring of the performance levels and exhaust emissions of agricultural tractors is expensive and time-consuming because it requires field measures. Thus, it is necessary to exploit new approaches that, employing the manufacturers' data and measured field data, enable the monitoring and evaluation of fuel consumption levels and pollutant emissions [18]. These parameters are determined based on the specific agricultural task carried out, the engine's working point, and the load connected to the power take-off (PTO) [19–21].

In the past decade, stringent emission standards have been introduced for non-road mobile machineries (NRMM), including agricultural tractors [22]. To comply with these regulations, tractor manufacturers have integrated technologies aimed at filtering particulate matter from the exhaust system, such as exhaust aftertreatment systems and particulate filters [23]. However, these approaches are characterized by high costs, significant spatial requirements, and an inability to completely mitigate the issue. Consequently, NRMM manufacturers are now exploring novel sustainable methodologies and technologies [24,25], aiming to curtail emissions and decrease fossil fuel consumption. The primary objective is the sustainable reduction of environmental impact, while simultaneously enhancing overall machine efficiency.

Among the sustainable technologies relevant to the agricultural sector, the development of hybrid electric tractors offers promising prospects. This approach could represent, in the future, the predominant direction for the development of hybrid NRMM drive systems [26–28]. The integration of a conventional ICE engine with an electric drive system is in line with the principles of sustainable agriculture, environmental preservation, and promotion of a greener food production.

Hybrid electric tractors can ensure the following advantages:

- Improved efficiency: The integration of an electric powertrain allows a more precise control of energy usage, optimizing the tractor's performance during farming operations;
- Fuel savings: Reduced reliance on fossil fuels is attained by harnessing electric power, leading to significant fuel savings and cost reductions;
- Lower emissions: Decreased emission of pollutant exhaust and GHGs during operations, contributing to a cleaner and more sustainable farming practice;
- Flexibility: The tractor can switch between the ICE and the electric power source, allowing the farmers to adapt their employment to different workloads and working conditions;
- Reduced noise and vibration: The electric motor operates quietly, reducing noise pollution in rural areas and improving the working environment;
- Decreased maintenance costs: The tractors have far lower maintenance requirements than their diesel counterparts because they have fewer mechanical parts, reducing the chances of their breaking down;
- Safety and stability: These tractors have centers of gravity positioned lower than those of their diesel counterparts, reducing the likelihood of their toppling or rolling over in uneven terrain.

The implementation of hybrid drive systems in agricultural tractors is in its initial phase; therefore, there are still a number of technological limitations. Solving these problems will enable the advent of mass-scale production of hybrid tractors [22]. Firstly, tractors are very versatile agricultural machines; they can perform a variety of operations, such as ploughing, soil tillage, fertilizing, and transport, demanding different levels of power and loads, so their power range is very wide, ranging from tens of kW up to hundreds of kW.

Furthermore, high technological and production costs, combined with the lack of efficient energy storage systems (ESSs), limit the spread of hybrid electric tractors [29]. The battery's energy density is 100 times less than the diesel's energy density [30]. This aspect implies that, in the future, agricultural tractors will still be powered by an ICE, although combined with an electric drive in a hybrid configuration.

The traditional methods of assessing tractor performance, fuel consumption, and pollutant emissions involve expensive and time-consuming field tests. Advanced technologies enable the replacement of some of these experiments with computer simulations, which assures the reliability of the results.

Therefore, the employment of simulation software in the modelling and design phase of an agricultural tractor can be very impactful [31]. Simulation tools enable the creation of digital twins of the vehicle, which represent the virtual prototype and replicate the behavior of the real machine [32]. Moreover, simulation models are very important, because they make it possible to test the machine under a variety of working conditions, looking in real-time at the effects of discrete design modifications in the powertrain configuration, avoiding the creation of expensive physical prototypes. One of the principles of creating numerical simulation models is to ensure that the model is as flexible and reusable as possible.

The use of simulation software in the agricultural sector to model and evaluate the performance of an agricultural tractor has not been yet reported. Moreover, in the scientific literature, there are not, at present, studies that evaluate and compare the performance levels and levels of pollutant emissions of "conventional" tractors, powered only by an ICE, and electric-hybrid tractors, which combine an ICE with an electric machine. The aim of this study is to analyze and assess the performance, as to $CO_2$ emissions and fuel consumption, of hybrid agricultural tractors and make a comparison with the "conventional" tractors, using a simulation software usually employed in the automotive sector.

Therefore, in this paper, the modeling and simulation of a two-wheel-drive agricultural tractor, in different configurations, during the execution of a custom-defined working cycle simulating the trailing, in the field, of a "big square baler" during the process of straw wrapping and baling, has been carried out [33,34]. The considered configurations were as follows: (i) the "conventional" configuration, that is, a tractor driven only by an ICE; (ii) the series electric-hybrid, which combines an ICE, a generator, and an electric motor; (iii) the parallel electric-hybrid, which combines an ICE and an electric motor. The performance, fuel consumption, $CO_2$ emissions, and depths-of-discharge of different hybrid electric tractors in varying configurations, including various forms of electric machine power, have been analyzed and compared to those of the "conventional" tractor. Furthermore, across multiple configurations, the context of hybrid-electric configurations has been especially studied.

Agricultural-tractor hybridization is still in its early stage, but it is a rapidly growing sector that will dominate the market in the near future, since it can ensure high performance, fuel savings, and low levels of pollutant emissions.

## 2. Materials and Methods

### 2.1. "Autonomie" Simulation Software

In the study described in this paper, the employed simulation software is "Autonomie" (https://www.anl.gov/taps/autonomie-vehicle-system-simulation-tool, accessed on 3 January 2024). It is a software developed by the Argonne National Laboratory Vehicle & Systems Mobility Group (VMS) for the modelling and simulation of vehicles [35]. It is used to assess the impact of a vehicle in terms of performance, energy and fuel consumption, emissions, and cost analysis. The "Autonomie" software was originally intended for simulating on-road vehicles, but, after introducing some modifications to the design of the powertrain and vehicle control systems, it can also be used to simulate the operation of NRMM, e.g., agricultural tractors [36]. The simulation approach is very useful for certain tasks: (i) easily evaluating the performance indicators in different working conditions;

(ii) comparing and sizing powertrain architectures; (iii) implementing new configurations without the need to construct any expensive prototypes [37].

The "Autonomie" software automatically builds and interconnects each subsystem included in the vehicle model. Once the user has configured the model with the definition of each single component and its parameters, an operating cycle must be selected. At this point, the software establishes the solver settings and initializes the model parameters, setting up the model, and then performs the simulation. In this paper, the models of a conventional tractor, a series electric-hybrid tractor, and a parallel electric-hybrid tractor have been developed by modifying some models of on-road vehicles already present in the software "Autonomie". Some parameters of the main blocks (i.e., ICE, electric motor, gearbox, and chassis) have been set according to the specifications of commercial tractors (e.g., the power and efficiency of electric motors and ICE, gearbox and final drive efficiency, weight of the vehicle, and wheel radius).

### 2.2. Working Cycle Definition

Some international standards, such as the US Federal Test Procedure (FTP) and the European Test Cycle (ECE), define the target speed profile to be employed for road vehicles; these are widely used in the automotive sector [38]. However, standardized working cycles for NRMM in the agricultural sector have not been yet defined; this prevents the utilization of a standard procedure to evaluate the sufficiency of the task's execution. Agricultural tractors differ from passenger cars in that the available power generated by the ICE is used not only for traction effort, but also to satisfy the large demand of power required by the implements connected to the PTO. It is worth noting that agricultural operations such as ploughing and tilling require high levels of torque at low speeds to "trail the load". The amount of power required strongly depends on the specific implement connected. To simulate agricultural operation, it is necessary to define a working cycle which imposes a speed profile with regard to time, but, in this case, is also important consider the power required by the implement connected to the PTO.

The simulated agricultural operation is the trailing of the big square baler HD 1270 (Cicoria Square Bales, Palazzo San Gervasio, Italy) by the New Holland 6090 (CNH Industrial, Torino, Italy) agricultural tractor (length, 5.3 m; width, 2.4 m; maximum power, 121 kW @2200 rpm; max torque, 710 Nm @1400 rpm) during the processes of straw wrapping and baling (Figure 1). Produced by "Cicoria Square Bales", the baler is 7 m long, has a width of 2.6 m, and weighs 7210 kg. It can be used to produce bales of straw with weights of 120 to 600 kg, or hay bales with weights from 200 to 900 kg [39]. The big square baler HD 1270T has been linked to the tractor by means of a universal joint OC-SJ316-50 produced by the Octis S.r.l company (Castiglione delle Stiviere, Italy), whose main technical characteristic was a max. torque of 2300 Nm.

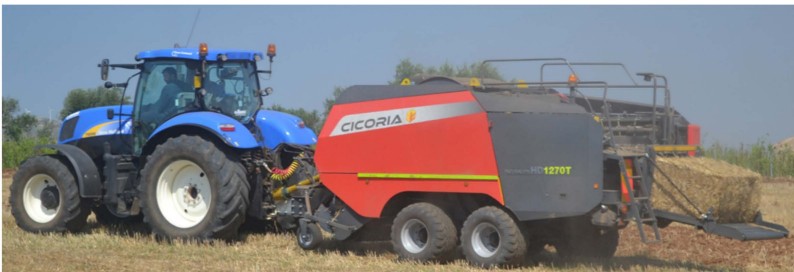

**Figure 1.** Big square baler (Cicoria HD 1270T) trailed by a tractor (New Holland 6090) during the tests. Source: Author's personal archives.

The custom-defined working cycle, presented in Figure 2, simulates the operation described immediately above, which has a duration of 1800 s. It involves several repetitions of straight sections and turning maneuvers, executed with constant speeds of 8 km/h and 4 km/h, respectively.

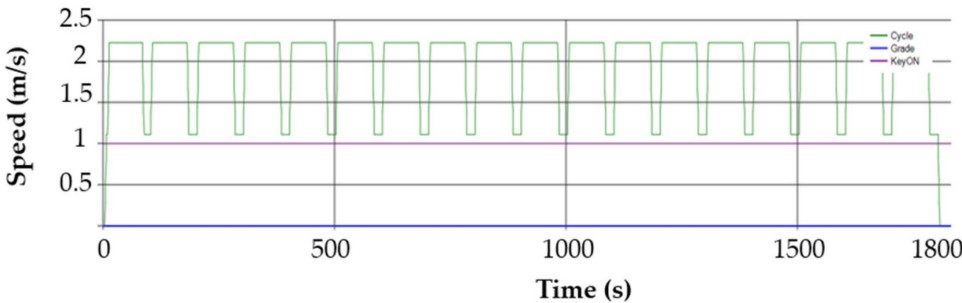

**Figure 2.** Custom-defined working cycle which simulates the trailing of the big square baler Cicoria HD 1270T. Source: Screenshot from the "Autonomie" software.

The Series PTO 420 sensor, produced by Datum Electronics (East Cowes, UK), is a non-contact drive-shaft torque-and-shaft power monitoring system. It has been employed to measure the torque required by the PTO and its angular velocity during the straw wrapping and baling processes [40,41]. This transducer uses a microprocessor circuit placed on the shaft to measure the deformation of its torque and rotational speed. Then, the data is sent to the stationary part and to the control unit, which is equipped with an RS232 connector enabling connection to a laptop [42,43]. Some technical features include the following: non-linearity, +/−0.1% FSD; uniqueness, +/−0.05% FSD; sampling frequency, 1–100 samples per second; output baud, 9600; and maximum torque, 1800 Nm. Figure 3 shows the Series PTO 420 rotary torque converter located between the tractor's power take-off and the Hooke joint.

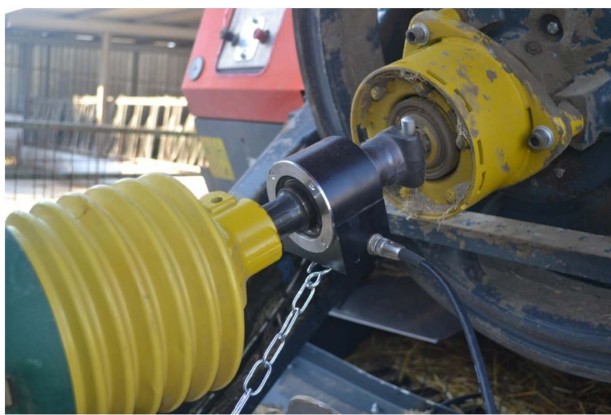

**Figure 3.** Contactless rotary torque transducer Datum Electronics Series PTO 420, linked to the tractor's PTO. Source: Author's personal archives.

### 2.3. Hybrid Electric Agricultural Tractor Powertrain

An agricultural tractor that is equipped with two different engines, one of which is an electric motor, is defined as a hybrid electric. Usually, this powertrain configuration has one bidirectional propulsion system and one either bidirectional or unidirectional propulsion system, as presented in Figure 4. In this way it is possible to recover part of the braking energy, which, in vehicles equipped with a conventional ICE, is usually dissipated in the form of heat [18,44,45]. Due to the way the two motors are connected, a distinction can be made between series and parallel electric-hybrid systems.

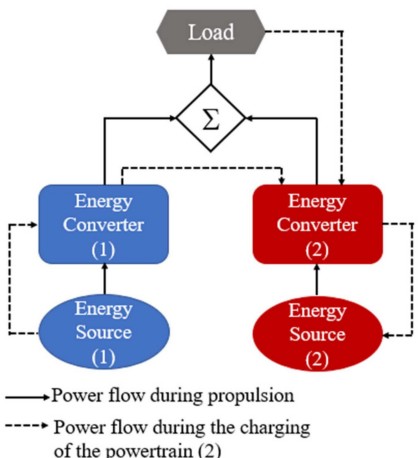

**Figure 4.** Hybrid electric agricultural tractor powertrain configuration. Source: Author's own elaboration.

In hybrid vehicles, the load power requirement (resistive power) is met by appropriately balancing the powers generated by the two engines (driving power), resulting in various operational configurations [46]: (*i*) the resistive power demand is solely satisfied by powertrain 1 or exclusively by powertrain 2, or simultaneously from both; (*ii*) powertrain 2 acquires power from the load, thereby effecting the regenerative braking; (*iii*) powertrain 2 receives power from powertrain 1; (*iv*) powertrain 2 simultaneously receives power from powertrain 1 and the load; (*v*) powertrain 1 concurrently supplies power to the load and powertrain 2; (*vi*) powertrain 1 provides power to powertrain 2, and the latter furnishes power to the load; or (*vii*) powertrain 1 supplies power to the load, and the load, in turn, provides power to powertrain 2.

The switching between these different operating configurations is managed by programmed electronic control units, which allows the system to optimize the vehicle's performance and efficiency.

### 2.4. Parameters of the Simulated Agricultural Tractors

In the study described in this paper, three custom-defined two-wheel-drive agricultural tractors of different sizes, namely, small (Tractor 1, 60 kW), medium (Tractor 2, 90 kW), and large (Tractor 3, 160 kW), have been modelled and simulated. Table 1 summarizes the defined models' parameters, which collectively resemble three commercial agricultural tractors. Each of the tractors have a direct-injection diesel engine with a 5-speed automatic transmission. The engine efficiency has been set to 40%, which is a standard value for a modern midsize diesel engine. The gearbox and the final drive efficiency has been set to 97%.

**Table 1.** Main parameters of the simulated models.

| Parameter | Tractor 1 | Tractor 2 | Tractor 3 |
|---|---|---|---|
| ICE Maximum Power @ 2200 rpm [kW] | 60 | 90 | 160 |
| Maximum Torque @ 1400 rpm [Nm] | 315 | 475 | 845 |
| Mass [kg] | 2050 | 5000 | 8000 |
| Wheel Radius [m] | 0.3 | 0.38 | 0.42 |

The friction force due to the air flow has been considered negligible due to the low speed at which the agricultural operation is taking place.

For Tractor 1 (60 kW), Tractor 2 (90 kW), and Tractor 3 (160 kW), several series and parallel electric-hybrid configurations have been modelled and simulated in the "Autonomie"

software. For the series powertrain, the different configurations have been obtained by varying the combination of the ICE–generator–electric-motor powertrain specifications, i.e., the power and size of the elements. Regarding the parallel powertrain, these have been found by varying the respective power levels of the ICE and the electric motor. The chosen configurations are those which satisfy the requirement of the target speed profile imposed by the working cycle.

Figure 5 shows the power (panel a) and torque (panel b) profiles as functions of the engine rotational speed, respectively, for an agricultural tractor powered by an ICE with a maximum power of 60 kW at 2200 rpm and a maximum torque of 315 Nm at 1400 rpm.

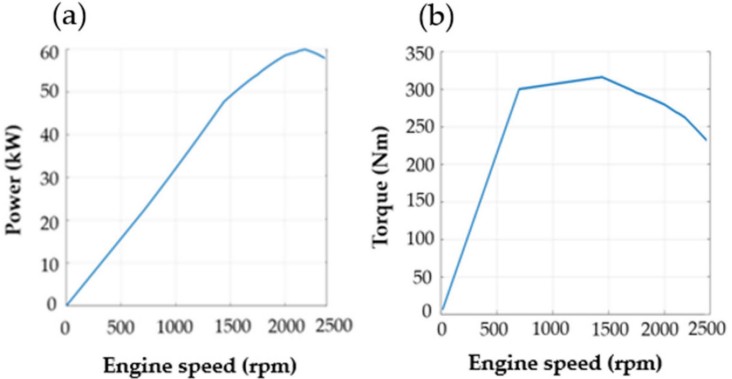

**Figure 5.** (**a**) ICE power profile vs. engine speed; (**b**) ICE torque profile vs. engine speed. Both are for an agricultural tractor powered by an ICE with maximum power of 60 kW at 2200 rpm and maximum torque of 315 Nm at 1400 rpm. Source: Author's own elaboration.

### 2.5. Simulink Models for Parameters Calculation

The "Autonomie" software is based on MATLAB and Simulink. The software interconnects the models of each subsystem, as developed in Simulink, constructing the entire vehicle's model. The Simulink models have been simulated and run at 100 Hz, using a fixed-step ode4 Runge–Kutta solver, with a time step of 0.01 s.

#### 2.5.1. Fuel Consumption Calculation

The Simulink model of the ICE is composed of four fundamental blocks: Engine Torque Calculation, Engine Thermal Calculation, Engine Fuel Consumption Calculation, and Engine Emissions. Focusing on the subsystem of Engine Fuel Consumption Calculation, as shown in Figure 6, it calculates the fuel consumption through a finite-state machine, the state of which depends on the fuel-rate map of the engine and on the state in which the ICE is present.

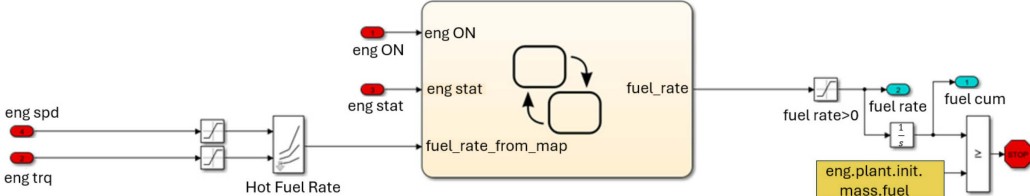

**Figure 6.** Simulink block for the fuel consumption calculation. Source: Screenshot from the "Autonomie" software.

If the ICE is started and the torque produced, ($T_{eng}$) is greater than the torque boundary of the fuel-rate map ($T_{min\ map}$), i.e., $T_{eng} > T_{min\ map}$; the instantaneous fuel consumption is a function of the engine power and the produced torque, and it depends on the fuel-rate map of the engine [47]. If the ICE is started and $T_{eng} < T_{min\ map}$, since no data are available in this region, the fuel consumption must be interpolated. When $T_{eng} = T_{min\ map}$, the fuel

rate at $T_{\text{min map}}$ is requested. The total mass of fuel used by the ICE during the defined working cycle is given by the Formula (1):

$$\text{Fuel consumption}_{\text{tot}} = \int (\text{Instantaneous fuel rate}) \, dt. \tag{1}$$

Starting from the fuel consumption and evaluating the performance levels of the ICE, i.e., its state and rotational speed, during the whole working cycle, the "Autonomie" software calculates the $CO_2$ emissions.

### 2.5.2. Battery State-of-Charge Calculation

The Simulink model of the plant of high-power batteries, employed as battery pack in the simulated series and parallel electric-hybrid configurations, is reported in Figure 7. It is composed of three main blocks: Voltage Calculation, Current Calculation, and State of Charge (SOC) Calculation. Focusing on the SOC calculation block, Figure 7 shows its subsystem [48]. The SOC is calculated by determining the variation of charge in the battery and dividing by the maximum capacity of the battery. The values of 0 and 1 are unattainable states, because they would represent the complete discharge and charge of the battery, resulting in a short life-cycle. In all the simulated configurations of the agricultural tractor, the initial SOC of the battery has been set to 70%.

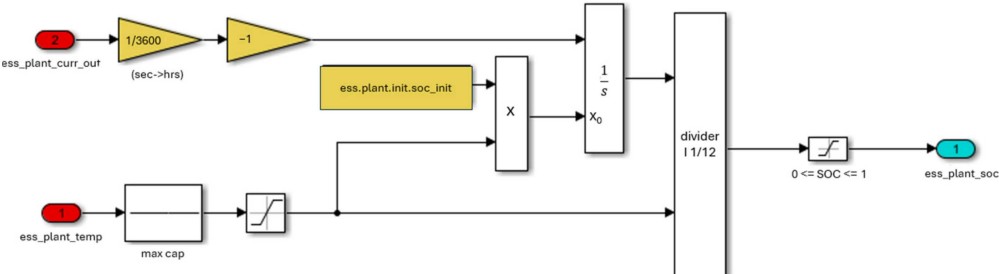

**Figure 7.** Simulink block for the SOC of battery calculation. Source: Screenshot from the "Autonomie" software.

Based on Figure 6, the *SOC* is given by Formula (2), where $SOC_{\text{init}}$ is the initial value of the *SOC* imposed, which is represented, in Figure 6, by the initial value of the integrator:

$$SOC = SOC_{init} + \Delta SOC, \tag{2}$$

The $\Delta SOC$ is calculated using Formula (3), where $I_{\text{in}}$ is the input current flowing in the battery from the bus and $C_{max}$ is the maximum charge capacity:

$$\Delta SOC = -\int \frac{I_{in}}{C_{max}} dt. \tag{3}$$

## 3. Results

### 3.1. Measuring the Torque and the PTO Angular Speed

The Datum Electronics PTO 420-series non-contact rotary torque monitoring system records data with intervals of 180 to 600 s, depending on the distance traveled by the agricultural tractor. Of the acquired data sets, one was arbitrarily selected for the analysis because the data sets did not show much diversity. Figure 8 shows the time function of the torque profile and the PTO angular velocity. Table 2 shows the basic descriptive statistics of the torque and the PTO angular velocity: minimum and maximum values, mean values, and standard deviations.

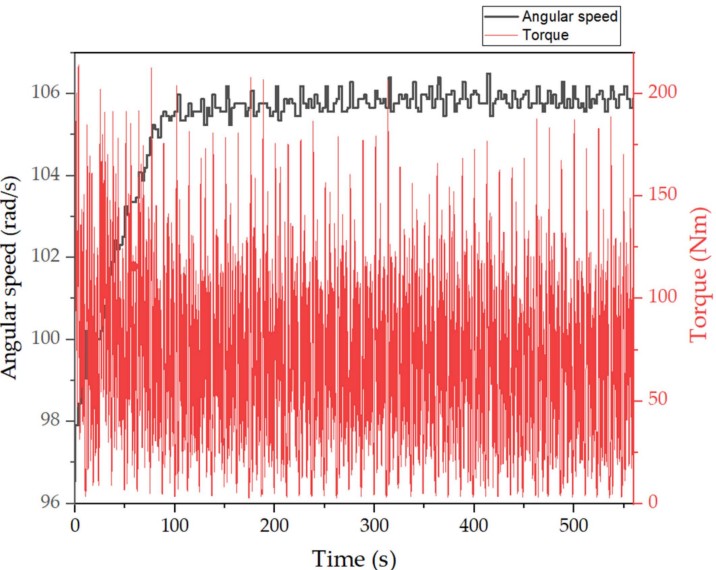

**Figure 8.** Time function of the torque profile and the PTO angular velocity. Source: Author's own elaboration.

**Table 2.** Basic descriptive statistics of torque and PTO angular velocity.

| | Min | Max | Average | Standard Deviation |
|---|---|---|---|---|
| PTO Angular Speed (rad/s) | 96.55 | 106.50 | 105.21 | 1.63 |
| Torque (Nm) | 2.64 | 214.41 | 69.25 | 40.94 |

Considering the mean PTO angular speed and torque ($T_{mean}$), it is possible to calculate the mean power ($P_{mean}$) required by the big square baler Cicoria HD 1270T during the straw wrapping and baling process; it is given by Formula (4).

$$P_{\text{mean } baler} = T_{mean} \times PTO\ Angular\ Speed_{mean} \sim 7.2\ \text{kW} \tag{4}$$

*3.2. Models' Simulations*

3.2.1. Conventional Agricultural Tractor

Figure 9 shows the model of a conventional agricultural tractor developed in the "Autonomie" software. The following basic blocks were used in the vehicle model: the driver; the environment model; the vehicle powertrain controller (VPC), which is the high-level controller; and the vehicle powertrain architecture (VPA). Due to the complexity of the powertrains, each architecture has its own system and subsystems tied to its own buses. At the most fundamental layer of abstraction, all models share an identical structure consisting of two automatically created blocks that allow the selection of inputs and the definition of units and data types, as well as a configurable block that is equivalent to a system installation. To adapt the predefined "Autonomie" vehicles' models to that of an agricultural tractor, several changes in different blocks of the model have been made. First of all, the Engine block has been modified, introducing a diesel engine with the parameters reported in Table 1; also, the Gearbox block and the Vehicle Dynamics block have been modified to adapt the model to the specifications defined in Table 1. Moreover, the big square baler HD 1270, which is the implement connected to the PTO, is represented by the block "Mechanical Accessory".

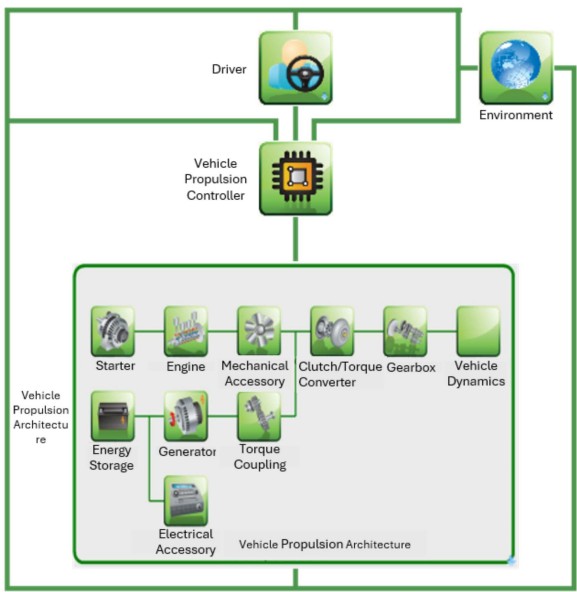

**Figure 9.** Model of a conventional agricultural tractor. Source: Screenshot from the "Autonomie" software.

Figure 10 shows, in blue, the desired speed profile imposed by the defined working cycle, and, in orange, the speed profile obtained from the model's simulation of (the conventional agricultural) Tractor 3 (160 kW).

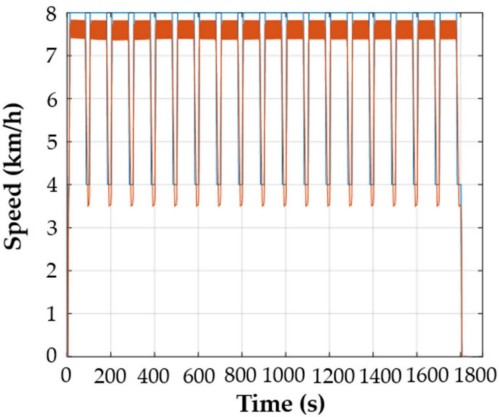

**Figure 10.** Speed profile imposed by the working cycle (blue line), and the speed profile followed by the model simulation (orange line), for Tractor 3 (160 kW). Source: Screenshot from the "Autonomie" software.

Table 3 summarizes the $CO_2$ emissions in kg/h and the fuel consumption in L/h for the three simulated conventional agricultural tractors.

**Table 3.** $CO_2$ emissions in kg/h and fuel consumption in L/h of the three simulated conventional agricultural tractors.

| Parameter | Conventional Tractor | | |
|---|---|---|---|
| | Tractor 1 (60 kW) | Tractor 2 (90 kW) | Tractor 3 (160 kW) |
| $CO_2$ Emission [kg/h] | 13 | 11 | 11.8 |
| Fuel consumption [L/h] | 5 | 4.7 | 5.8 |

### 3.2.2. Series Electric-Hybrid Agricultural Tractor

The model of a series electric-hybrid agricultural tractor, as developed in the "Autonomie" software, is presented in Figure 11. In the series electric-hybrid configuration, an electric generator is coupled with the ICE, resulting in a diesel generator set (GENSET). In this way, the ICE is only used to make up for the energy shortage of the batteries. This is a technology in which the ICE is used only to replenish the energy deficiency in the batteries. The inefficiency of electric machines causes large energy losses, and this is due to the fact that, in this technology, the power available for traction, PTO, and auxiliary devices comes entirely from the process of double energy conversion: in the generator—from mechanical energy into electrical energy; and then in the traction motor—again from electrical energy to mechanical energy. In this configuration, the ICE engine is completely decoupled from the wheels, so it only operates on the brake-specific fuel consumption (BSFC) basis, which results in fuel savings. The main disadvantage of the series architecture is the use of two electrical machines, a functional generator and a traction motor, which makes the vehicle heavier, larger, and more expensive. The introduction of ESS as a means of storage of large amounts of electricity ensures an appropriate energy autonomy, equalization of energy peaks, reduction of the size of the ICE, and faster activation of energy reserves.

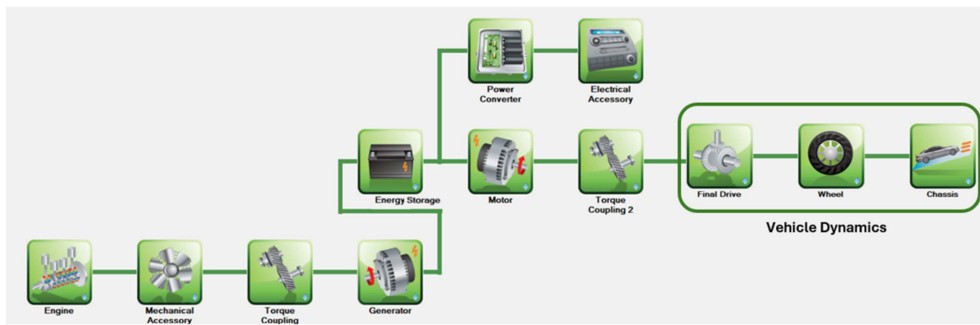

**Figure 11.** Model of a series electric-hybrid agricultural tractor. Each block that can be navigated into has a small down blue arrow on the bottom right of their picture. Source: Screenshot from the "Autonomie" software.

As for the series electric-hybrid powertrains, for each tractor, two different combinations (Config. A and Config. B) of the ICE–generator–electric-motor configuration have been simulated and analyzed. The electric-motor efficiency has been set to 90%. The chosen powertrain configurations respect the speed profile imposed by the working cycle. Table 4 reports the data from the simulated series electric-hybrid configurations.

**Table 4.** Simulated series electric-hybrid configurations.

| Parameter | Series Electric-Hybrid Tractor | | | | | |
|---|---|---|---|---|---|---|
| | Tractor 1 (60 kW) | | Tractor 2 (90 kW) | | Tractor 3 (160 kW) | |
| | Config. A | Config. B | Config. A | Config. B | Config. A | Config. B |
| ICE [kW] | 50 | 45 | 70 | 60 | 110 | 100 |
| Generator [kW] | 5 | 5 | 10 | 10 | 15 | 15 |
| Electric Motor [kW] | 5 | 10 | 10 | 20 | 35 | 45 |

Table 5 reports the $CO_2$ emissions in kg/h, the fuel consumption in L/h, and the depth of discharge of the battery pack of the simulated series electric-hybrid agricultural tractor; note that that the initial SOC has been set to 70%.

**Table 5.** $CO_2$ emissions in kg/h, fuel consumption in L/h, and depth of discharge of the simulated series electric-hybrid agricultural tractor.

| Parameter | Series Electric-Hybrid Tractor | | | | | |
|---|---|---|---|---|---|---|
| | Tractor 1 (60 kW) | | Tractor 2 (90 kW) | | Tractor 3 (160 kW) | |
| | Config. A | Config. B | Config. A | Config. B | Config. A | Config. B |
| $CO_2$ emission [kg/h] | 1.1 | 0.7 | 3 | 2.5 | 4.6 | 3.7 |
| Fuel Consumption [L/h] | 0.5 | 0.4 | 1.4 | 1.2 | 2.2 | 1.8 |
| $\Delta$ SOC [%] | $-22.5$ | 24.5 | $-25.1$ | $-26.6$ | $-27.1$ | $-27.7$ |

### 3.2.3. Parallel Electric-Hybrid Agricultural Tractor

Figure 12 shows the model of a parallel electric-hybrid agricultural tractor developed in the "Autonomie" software, in which the electric motor is mechanically coupled with the ICE and the final drive through the gearbox. A clutch is placed between the ICE and the electric motor to decouple the two motors, allowing the tractor to operate in a fully electric mode. This architecture ensures that the power from the ICE can be transferred mechanically to the wheels, as in a conventional agricultural tractor, and the electric motor is used for the overall support of the ICE. During low power operations, the ICE can be used as a generator to recharge the ESS. This architecture is most common in hybrid vehicles, because it requires the addition of one electric machine, reducing the size of the ICE without having to completely change the design.

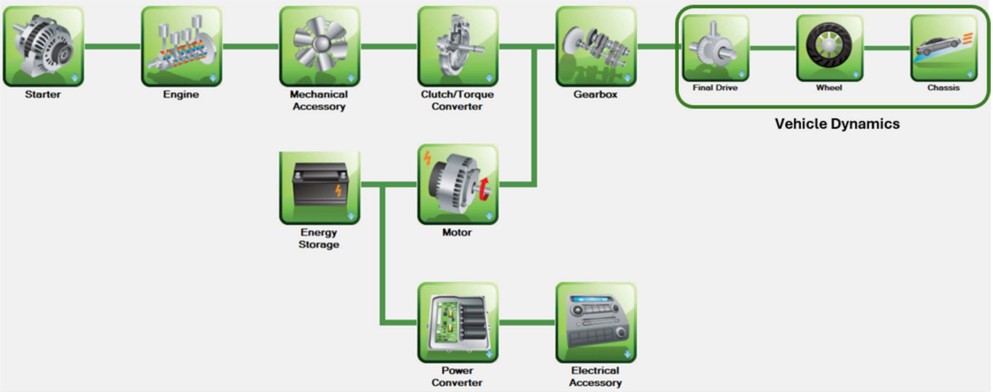

**Figure 12.** Model of a parallel electric-hybrid agricultural tractor. Each block that can be navigated into has a small down blue arrow on the bottom right of their picture. Source: Screenshot from the "Autonomie" software.

As for the parallel electric-hybrid powertrains, for each tractor, two different combinations (Config. A and Config. B) of the ICE–electric motor configuration have been simulated and analyzed. The electric-motor efficiency has been set to 90%. The chosen powertrain configurations respect the speed profile imposed by the working cycle. Table 6 reports the data from the simulated parallel electric-hybrid configurations.

**Table 6.** Simulated parallel electric-hybrid configurations.

| Parameter | Parallel Electric-Hybrid Tractor | | | | | |
|---|---|---|---|---|---|---|
| | Tractor 1 (60 kW) | | Tractor 2 (90 kW) | | Tractor 3 (160 kW) | |
| | Config. A | Config. B | Config. A | Config. B | Config. A | Config. B |
| ICE [kW] | 50 | 45 | 80 | 75 | 140 | 110 |
| Electric Motor [kW] | 10 | 15 | 10 | 15 | 20 | 50 |

Table 7 summarizes the $CO_2$ emissions in kg/h, the fuel consumption in L/h, and the depth of discharge of the battery pack of the simulated parallel electric-hybrid agricultural tractor.

**Table 7.** $CO_2$ emissions in kg/h, fuel consumption in L/h, and depth of discharge of the simulated parallel electric-hybrid agricultural tractor.

| Parameter | Series Electric-Hybrid Tractor | | | | | |
|---|---|---|---|---|---|---|
| | Tractor 1 (60 kW) | | Tractor 2 (90 kW) | | Tractor 3 (160 kW) | |
| | Config. A | Config. B | Config. A | Config. B | Config. A | Config. B |
| $CO_2$ emission [kg/h] | 2.3 | 1.6 | 4.5 | 4.2 | 7.7 | 6.3 |
| Fuel Consumption [L/h] | 1.1 | 0.8 | 2.1 | 2 | 3.5 | 3 |
| $\Delta$ SOC [%] | −3.3 | −5 | −5.5 | −6.4 | −7 | −7.9 |

## 4. Discussion

### 4.1. Conventional Powertrain

As for the conventional agricultural tractor, Table 3 reports the $CO_2$ emissions generated by the vehicle and the fuel consumption for the three simulated tractors, i.e., Tractor 1 (60 kW), Tractor 2 (90 kW), and Tractor 3 (160 kW). Within Table 3, it is possible to highlight that the fuel consumption is similar for the three considered configurations, and that it is equal to 5 L/h, 4.7 L/h, and 5.8 L/h, respectively. This mainly depends on the operating conditions of the ICE, and not strictly on the power of the tractor. Since the fuel consumptions are comparable for the three tractors, the $CO_2$ emissions are also similar, and stand at about 13 kg/h, 11 kg/h, and 11.8 kg/h, respectively.

### 4.2. Series Electric-Hybrid Powertrain

Table 5 points out that the $CO_2$ emissions and fuel consumption of a series electric-hybrid agricultural tractor are, relevantly, lower than the ones generated by the conventional tractor; for instance, for Tractor 1, they are ten times lower. The configurations in which the electric motor is more powerful (Config. B for each tractor) have lower $CO_2$ emissions and fuel consumption, compared to Config. A. When increasing the power of the tractor, as in varying from Tractor 1 to Tractor 3, for instance, the fuel consumption increases, as well as the $CO_2$ emissions, as expected. The depth of discharge of the battery pack stands at 23.5%, on average, for Tractor 1; 25.9%, on average, for Tractor 2; and 27.5%, on average, for Tractor 3.

### 4.3. Parallel Electric-Hybrid

Table 7 shows the simulation results for a parallel electric-hybrid agricultural tractor. It was noticed that the $CO_2$ emissions and fuel consumption of Tractor 1 are significantly lower compared to the conventional tractor, they are five times lower. In configurations with higher power electric motors (i.e., Config. B for each tractor), lower $CO_2$ emissions and fuel consumption were observed compared to the Config. A. When increasing tractor power, for example from Tractor 1 to Tractor 3, fuel consumption and $CO_2$ emissions increase as expected. Moreover, the battery discharge depth is on average 4.1% for Tractor 1, 6% for Tractor 2 and 7.5% for Tractor 3.

### 4.4. General Evaluation

Tables 4 and 6 highlight that it is possible to downsize the ICE of an agricultural tractor in association with the introduction of one, in the case of parallel electric-hybrid architecture, or two, in the case of series electric-hybrid architecture, electric machines. Moreover, the electric motors allow a more flexible speed control, which permits the following of the target speed profile of the working cycle in a more precise way, as compared to the conventional tractor, as shown in Figure 13. Each of these electric machines can be independently operated to ensure that it consumes only the power required for its operation.

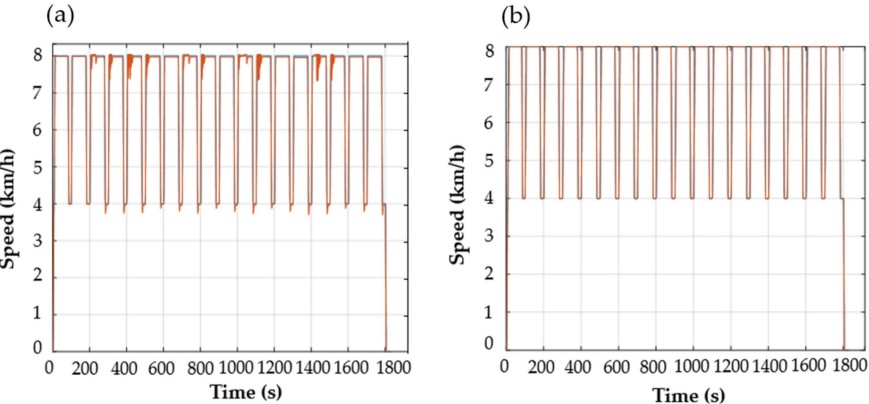

**Figure 13.** Speed profile imposed by the working cycle (blue line) and speed profile followed by the model simulation (orange line) of Tractor 3 (150 kW) in (**a**) parallel electric-hybrid configuration and (**b**) series electric-hybrid configuration. Source: Screenshot from the "Autonomie" software.

Using Tables 5 and 7, it is possible to compare the $CO_2$ emissions, the fuel consumption, and the battery pack discharge of the two electric-hybrid configurations. The series electric-hybrid architecture uses the electric motor as main traction powertrain and the ICE only to compensate for the energy shortage in the batteries, and thus, the $CO_2$ emissions and the fuel consumption are lower than those of the parallel electric-hybrid architecture. Furthermore, the series architecture, requiring more electric energy, has a faster discharge of the battery pack, as expected.

However, the series architecture requires two electric machines; thus, this configuration is more difficult to utilize in the hybridization of existing vehicles. Indeed, the most employed configuration on the market is the parallel architecture.

## 5. Conclusions

This paper presents the modeling and the simulation of three differently sized (60 kW (small), 90 kW (medium), and 150 kW (large)) agricultural tractors, whose models' parameters resemble three commercial tractors, during the execution of a task which simulates the trailing in the field of the big square baler HD 1270T during a straw wrapping and baling process. The modelling and the simulations were performed using the "Autonomie" simulation software, using as a starting point some models of on-road vehicles already present in the software. The three different analyzed configurations were: (i) the conventional one, characterized only by the ICE; (ii) the series electric-hybrid, which includes the ICE and two electric machines, i.e., a generator and an electric motor; and (iii) the parallel electric-hybrid, which is composed of the ICE and an electric motor.

The analysis and the evaluation of the performance levels of the task execution, the $CO_2$ emissions, and the fuel consumption associated with the different configurations have been carried out. A detailed study regarding these electric hybrid configurations has also been carried out to compare the depth of discharge of the battery pack. Moreover, a comparison between two different configurations of each hybrid powertrain, varying the power specifications of the electric machines for the series and the parallel architecture, has been performed to investigate if it is possible to downsize the ICE while maintaining the same performance levels during the execution of the task.

The simulation results highlight that the levels of fuel consumption and $CO_2$ emissions of the series and parallel electric-hybrid configurations are, relevantly, lower than the ones generated by the conventional tractor, in particular, ten times and five times, respectively. Config. B, which in both hybrid architectures has the more powerful electric motor, has lower levels of fuel consumption and $CO_2$ emissions with regard to Config. A. When increasing the power of the ICE of the tractor, varying from Tractor 1 to Tractor 3, for instance, both the fuel consumption and the $CO_2$ emissions increase, as expected.

The series electric-hybrid architecture, using the electric motor as the main traction powertrain and the ICE as only a booster, shows lower levels of fuel consumption and $CO_2$ emissions with respect to the parallel architecture, but has a quicker discharge of the battery pack, as expected. However, the series architecture is not frequently employed in commercial hybrid electric vehicles, because it requires two electric machines, increasing the weight and the complexity of the vehicle.

This study highlights the fact that the hybridization of agricultural tractor powertrains is a sustainable approach to reduce pollutant emissions and fuel consumption. The simulation results show clearly that hybridization cuts down on the environmental impact connected to the employment of agricultural tractors and other farming machines in agricultural operations. Hybridization represents one of the most impactful technologies for the development of greener and more sustainable farming machines.

**Author Contributions:** Conceptualization, F.P. and S.P.; methodology, F.P. and K.Ł.-W.; formal analysis, F.P. and K.G.; investigation F.P., K.Ł.-W., K.G. and S.P.; data curation, F.P. and K.G.; writing—original draft preparation, F.P. and S.P.; writing—review and editing, F.P., K.Ł.-W., K.G. and S.P.; supervision, F.P. and S.P. All authors have read and agreed to the published version of the manuscript.

**Funding:** This research received no external funding.

**Institutional Review Board Statement:** Not applicable.

**Informed Consent Statement:** Not applicable.

**Data Availability Statement:** The data presented in this study are available on request from the corresponding author. The data are not publicly available due to software limitations.

**Conflicts of Interest:** The authors declare no conflicts of interest.

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
