# Peer review of "Sustainability Evaluation of Hybrid Agriculture-Tractor Powertrains"

_sustainability, doi:10.3390/su16031184_

Round 1

Reviewer 1 Report

Comments and Suggestions for Authors

General comment:

This paper investigated the modeling and the simulation conducted using a simulation software typically used for on-road vehicles, of a two wheel-drive agricultural tractor in three different configurations. The evaluation and the comparison of the fuel consumption, CO2 emissions and the depth of discharge of the different configurations were carried out to determine if it is possible to downsize the ICE maintaining the same performances. The authors pointed out that the hybridization of agricultural tractors powertrains was one of the most promising approaches to reduce pollutant emissions and fuel consumption.

In this paper, authors explained the background of this study, simulation method, and analysis procedure well. And simulation results and discussion may be acceptable for publication. However, there were lots of mistakes in the sentences, and it was also found simulation results to be discussed. Reviewer does not recommend this paper for publication with this status. The author should address the following issues.

Major issue 1:

In line 81, authors explained “Less Maintenance Costs” as a merit of hybrid electric tractors. Reviewer does not agree this content because the hybrid electric tractors have more parts compared to the conventional engines. Please give author’s opinion on this or make corrections.

Major Issue 2:

Have you considered the efficiency and losses between blocks in the model simulation?  If so, please provide relevant evidence.

Minor Issues: please address the following things.

-          The font sizes of the legend in Figs. 2, 5, 6, 7, 9, 11and 12 are too small. Please make it larger so readers can read it.

-          At Eq.(1), Instantaneous fuel rate should be indicated by putting it in parentheses. Ex) (Instantaneous fuel rate)

-          At Eq.(3), the integral expression are not correctly displayed. Please indicate correctly the relationships of the input current and the maximum charge capacity within integral expression.

-          At Table 1, PTO should be corrected to “PTO Angular Speed”.

-          At Eq.(4), PTO should be also corrected to “PTO Angular Speed”.

-          At Table 2, the unit of power, kW, written after 60, 90, and 160 should be omitted.

-          In Fig. 8, the x-axis title, Angular Speed should be changed to “Engine speed”.

-          The power of tractor 3 in line 319 and in the title of Fig. 10 is not match. Please correct it.

Author Response

Dear Reviewer, 
We would like to extend our sincere gratitude for the time and expertise you dedicated to reviewing our paper. Your comprehensive comments and constructive feedback have been instrumental in significantly improving the overall quality of our manuscript.

In the attached document, you will find detailed responses addressing each of your comments. We have not only incorporated your valuable suggestions but have also taken into account recommendations obtained from all three reviewers. This collaborative effort has been crucial in refining the paper and ensuring its accuracy and clarity.

Thank you for your time, dedication, and commitment to maintaining the rigor of academic publications.

With best regards,
Authors

Reviewer 2 Report

Comments and Suggestions for Authors

The manuscript made evaluation of hybrid agriculture tractors by using a simulation software typically used for on-road vehicles. There are the following problems:

(1) Figure 1 and Figure 3 are photographs of real object. Could you add some geometric parameter photos? Photographs of real objects look unprofessional in an academic paper.

(2) In this manuscript, the employed simulation software is “Autonomie”. Is there any experimental data to compare with the results of the numerical calculations, and how do you prove the accuracy of the numerical calculations? The is the key problem in the manuscript.

(3) The research data in the results appear to be less, could you add some data figures such as Figure 10 and Figure 13?

(4) The size of texts in Figure 5, Figure 6 and Figure 11 is too small, you should improve the quality of the figures.

(5) The structure of this paper is similar to that of a literature papery, and the discussion in Part IV can be placed into the results of Part III.

(6) The space between the headers and the Tables in the manuscript is too large, especially in Table 1.

Comments on the Quality of English Language

90%

Author Response

(The authors gave the same response as above.)

Reviewer 3 Report

Comments and Suggestions for Authors

General comment:

The peer-reviewed manuscript is the authors original, innovative research work. The paper has scientific and social justification. The paper has minor elements of repetition/plagiarism. For the most part, the manuscript is well-written. Some chapters need to be amended and supplemented. I propose that the manuscript be published when the following changes and additions are made to it.

Comment 1:

Line 15: It is unclear what they are polluting. It should be said that it refers to soil and air pollution.

Comment 2:

Lines 34-35: Unclear sentence. In relation to which machines is meant?

Comment 3:

Line 36: Display number 2 using the “Subscript” option. Also, these minor corrections should be made in several places in the work.

Comment 4:

Line 40: Scientific papers should not be written in possessive adjectives: (our knowledge). Also, the same term that needs to be corrected is repeated in several places (Lines 112 and 115).

Comment 5:

Lines 50-51: This sentence should be retold because of plagiarism.

Comment 6:

Line 51: When the abbreviation PTO is introduced for the first time, it should be preceded by the full name.

Comment 7:

Line 65-67: This sentence should be retold because of plagiarism.

Comment 8:

Line 122: Also, scientific papers should not be written in possessive pronouns: (we have). Write the paper in the third person and past tense.

Comment 9:

The introductory chapter is well written. The introductory chapter needs to be expanded in terms of the impact on the environment, that is, to highlight the lack of internal combustion engines, i.e. their consumption not only of fuel but also of oil. In practice, they occur in cases where, due to improper operation of the engine, oil leaks on the seals. For more details, the authors can consult the following research: “Examination of Motor-Oils in Exploitation at Agricultural Tractors in Process of Basic Treatment of Plot, 19(2), 2013, pp.314-322“. All moving parts of the engine, such as piston rings, valve guides, camshafts, etc., are to a certain extent under the influence of friction. Frictional energy dissipation contributes significantly to mechanical losses, which account for 10–15% of the energy generated by an internal combustion engine. For more details, the authors can consult the following research: Impact of Diagnostics State Model to the Reliability of Motor Vehicles, 21(2), 2015, pp.511-522.

Comment 10:

Line 168: Which tractor is it about? It is necessary to specify the manufacturer and model, i.e. its power.

Comment 11:

Line 175: Time work cycle of 1800 s, it would be better to present it in minutes for the sake of seeing the real picture and facilitating the understanding of the problem for future readers.

Comment 12:

In chapter 2.2. more information about the used cardan shaft should be provided (manufacturer, type, max. power or torque).

Comment 13:

Lines 186-187: This sentence  - was probably taken from the catalog? The sentence instructs the users how to do it. Delete this sentence.

Comment 14:

Lines 215-217: The text is unreasonable. It is necessary to clarify and simplify the sentence.

Comment 15:

Figure 5 is of poor quality. The text on the image is illegible.

Comment 16:

Line 264: It should be said: Table 1 shows.....

Comment 17:

The material and methods of operation is not well described. Methodological descriptions are mentioned for the first time in the Research Results. That is, in chapter 3.2. It is said that the parameters of the model have been tested on 3 agricultural tractors. These data should be given and described in the chapter on material and method of operation.

Comment 18:

Line 274: It is necessary to provide more information about the tractors themselves (manufacturer, type and power).

Comment 19:

Figure 9 is of poor quality. The text on the image is illegible.

Comment 20:

Figure 11 part of the text is not visible.

Comment 21:

Lines 282-390: This data should be given in the form of a Table.

Comment 22:

Line 449: Delete the beginning of the sentence: (To conclude).

Comment 23:

The research results chapter is fundamentally well presented.

Comment 24:

The choice of reference is good. Introduce suggested references describing the disadvantages of internal combustion engines.

Author Response

(The authors gave the same response as above.)

Reviewer 4 Report

Comments and Suggestions for Authors

This paper conducts an analysis of electric hybrid tractors, focusing on their performance, fuel efficiency, emission levels, and battery discharge characteristics, utilizing advanced modeling and simulation methodologies. A comparative study is presented, juxtaposing the outcomes of these hybrid tractors with those of traditional Internal Combustion Engine (ICE) tractors. This paper needs some improvements to be considered for publication. First of all to improve the abstract, it is essential to clearly articulate these significant findings, highlighting the quantitative and qualitative advancements over conventional tractors. Moreover, the conclusion section should be meticulously crafted to succinctly summarize these critical results, providing more than a generic statement on emissions and fuel consumption reductions. It should encapsulate the key takeaways, offering a clear understanding of the study's implications in the broader context of agricultural technology and sustainability.

Some detail comments to follow up as below:

-          Ln. 137: Can the author describe Autonomie’s modeling and how it simulates the vehicle to provide performance and emissions data. Ln.151 and section 3.3.1 only discuss it in general black box terms without knowing the fundamental models of it.

-          Ln. 218: What is the relation between Autonomie and Simulink models?

-          Ln. 268, Ln. 296: What does it mean own elaboration? Are the data made up?

-          Ln.396: Can the author show graphical comparisons between the cases of conventional and hybrid ICE in terms of the performance, fuel efficiency, and emission levels.

Author Response

Dear Reviewer,

We sincerely appreciate your time and expertise in reviewing our paper. Your comprehensive comments and constructive feedback have significantly improved the overall quality of our manuscript.

Thank you for your dedication to maintaining the rigor of academic publications.

Best regards,
Authors

Round 2

Reviewer 2 Report

Comments and Suggestions for Authors

It's OK.

Author Response

(The authors gave the same response as above.)

Reviewer 3 Report

Comments and Suggestions for Authors

The manuscript has been significantly revised and improved. I propose to publish the manuscript in this version.

Author Response

(The authors gave the same response as above.)

Reviewer 4 Report

Comments and Suggestions for Authors

Thank you for following the suggestions. I am satisfied with the improvements made and can recommend for publication.